# Hesperidin, Hesperetin, Rutinose, and Rhamnose Act as Skin Anti-Aging Agents

**DOI:** 10.3390/molecules28041728

**Published:** 2023-02-11

**Authors:** Renáta Novotná, Denisa Škařupová, Jiří Hanyk, Jitka Ulrichová, Vladimír Křen, Pavla Bojarová, Katerina Brodsky, Jitka Vostálová, Jana Franková

**Affiliations:** 1Department of Medical Chemistry and Biochemistry, Faculty of Medicine and Dentistry, Palacký University Olomouc, Hněvotínská 3, 775 15 Olomouc, Czech Republic; 2Laboratory of Biotransformation, Institute of Microbiology, Academy of Sciences of the Czech Republic, Vídeňská 1083, 142 00 Prague 4, Czech Republic; 3Department of Biochemistry and Microbiology, University of Chemistry and Technology Prague, Technická 3, 166 28 Prague 6, Czech Republic

**Keywords:** skin aging, hesperetin, hesperidin, rutinose, rhamnose, normal human dermal fibroblast

## Abstract

Aging is a complex physiological process that can be accelerated by chemical (high blood glucose levels) or physical (solar exposure) factors. It is accompanied by the accumulation of altered molecules in the human body. The accumulation of oxidatively modified and glycated proteins is associated with inflammation and the progression of chronic diseases (aging). The use of antiglycating agents is one of the recent approaches in the preventive strategy of aging and natural compounds seem to be promising candidates. Our study focused on the anti-aging effect of the flavonoid hesperetin, its glycoside hesperidin and its carbohydrate moieties rutinose and rhamnose on young and physiologically aged normal human dermal fibroblasts (NHDFs). The anti-aging activity of the test compounds was evaluated by measuring matrix metalloproteinases (MMPs) and inflammatory interleukins by ELISA. The modulation of elastase, hyaluronidase, and collagenase activity by the tested substances was evaluated spectrophotometrically by tube tests. Rutinose and rhamnose inhibited the activity of pure elastase, hyaluronidase, and collagenase. Hesperidin and hesperetin inhibited elastase and hyaluronidase activity. In skin aging models, MMP-1 and MMP-2 levels were reduced after application of all tested substances. Collagen I production was increased after the application of rhamnose and rutinose.

## 1. Introduction

Population aging creates socioeconomic problems related to the quality of life. The aging of connective tissue is visible on the skin in the form of wrinkles and fibrotization of the skin, but, most importantly, it affects the flexibility of the musculoskeletal system (tendons, ligaments) and the flexibility of the veins. Aging also seems to affect all skin layers. Skin aging is a complex progressive process leading to functional and esthetic changes in the skin tissue [1]. The aged epidermis shows a reduced capacity for barrier function and recovery after damage [2]. In addition to the epidermis, the epidermal-dermal junction and the dermis are also getting thinner. The dermal extracellular matrix (ECM) shows structural changes, including the degradation of collagen and elastin fibers as well as hyaluronan degradation. A decrease in fibroblasts also contributes to the changes and degradation of the ECM, resulting in progressive thinning of the skin, increased wrinkling, and loss of elasticity [3]. These changes in skin tissue are the result of the action of various internal and external factors [4].

One of the internal factors responsible for the stimulation of aging is the nonenzymatic glycation of proteins and the associated formation of advanced glycated end products (AGEs). These molecules cause oxidative stress and damage to vascular endothelial/smooth muscle/connective tissue-cell and renal cell damage in AGEs-related diseases (such as physiological aging and also neurodegenerative diseases, diabetes mellitus and its complications) [5,6]. AGEs modify skin proteins (such as collagen, elastin, fibronectin, and laminin) and alter their functions/properties [7]. Therefore, it is useful to reduce the formation of AGEs, and the use of natural substances is one of the possible preventive strategies.

The flavonoid hesperidin is known for its anti-oxidant, chelating, and anti-aging properties [8]. It has also been shown to act as an effective anti-photoaging factor by regulating the activation of matrix metalloproteinases (MMPs) and the production of pro-inflammatory interleukins (ILs) [9]. Hesperidin is a flavanone glycoside (rutinoside; 6-*O*-α-l-rhamnopyranosyl-β-d-glucopyranoside) (Figure 1.) that, together with its aglycone hesperetin, is mainly found in bitter oranges (*Citrus aurantium*), in peppermint and some other plants. Hesperetin is also involved in increasing the hydration of the *stratum corneum* as well as increasing the amount of hyaluronic acid (HA) in the dermis [10]. Transdermal distribution of hesperidin has also been demonstrated. The results of the study by Hering et al. (2021) showed that hesperidin was able to cross the *stratum corneum*, the upper layer of the epidermis, and penetrate the deeper layers of the epidermis and dermis [11]. Although the bioavailability of hesperidin and hesperetin is relatively low, it can be increased by modifying a number of substances for topical application. The skin microbiome also could play an important role in the bioavailability of hesperidin [12].

A beneficial effect of rhamnose, a monosaccharide contained in rutinose, on the morphology of the reconstructed skin has been demonstrated by an increase in epidermal thickness and procollagen I production [13]. Additionally, rhamnose-rich polysaccharides and oligosaccharides inhibit the cytotoxic effects of AGEs [14,15]. These findings led us to study the effect of the monosaccharide rhamnose and compare it with that of the disaccharide rutinose.

In our study, we focused on the effect and comparison of the efficacy of hesperidin, hesperetin, rutinose, and rhamnose as potential anti-aging agents in vitro on NHDFs cultivated in medium with (1) low glucose and (2) high glucose and (3) medium supplemented with AGEs. Their ability to inhibit the activities of enzymes associated with skin aging, particularly hyaluronidase, collagenase, and elastase, which remodel the components of the extracellular matrix, was investigated by tube tests. Since senescent cells overexpress a number of molecules, including inflammatory cytokines (IL-6, IL-8) and MMP-1 (collagenase), MMP-2 (gelatinase A) [16], the effects of the investigated compounds on the above-mentioned markers were evaluated. The production of pro-collagen I in NHDFs was also determined after treatment with hesperidin, hesperetin, rutinose, and rhamnose in individual skin-aging models in vitro [17].

## 2. Results

### 2.1. Effect of Studied Compounds on the Activity of Enzymes Associated with Skin Aging (Tube Tests)

Modulation of elastase, hyaluronidase, and collagenase activity by rhamnose, rutinose, hesperidin, and hesperetin was evaluated by in vitro spectrophotometric and spectrofluorometric tube methods. The efficacy of rhamnose and rutinose was evaluated in a concentration range of 0.25–100 mM for elastase and hyaluronidase, and 10–100 mM for collagenase. The efficacy of hesperidin and hesperetin was evaluated in a concentration range of 0.5–100 µM for elastase and hyaluronidase.

In the tube tests, rutinose proved to be a more effective inhibitor of elastase, hyaluronidase, and collagenase activity than rhamnose. The inhibition by rutinose was dose-dependent. At the minimum concentration used (0.25 mM), it significantly inhibited elastase activity by 22%. At the maximum concentration used, rutinose inhibition reached the efficacy of the benchmark elastase inhibitor oleanolic acid (Figure 2A).

Rhamnose showed inhibition of hyaluronidase in the concentration range of 0.25–100 mM, reaching 60–70% inhibition (Figure 2C). Elastase was inhibited by rhamnose in the concentration range of 1–100 mM; 1 mM rhamnose significantly inhibited elastase activity by 23% (Figure 2A). Weak inhibition of collagenase activity (ca. 15%) was also observed at higher concentrations of rhamnose (25–100 mM; Figure 2E). Similar to rhamnose, hesperidin, and hesperetin showed the greatest effect in inhibiting hyaluronidase. In the concentration range of 0.5–100 µM, they inhibited the activity by 60–70% (Figure 2D). They also weakly inhibited elastase activity in a dose-dependent manner (in the concentration range 1–100 µM, Figure 2B), with hesperidin having a slightly stronger effect than hesperetin. Their effect on the inhibition of collagenase activity was not established.

### 2.2. Effect of Hesperidin, Hesperetin, Rutinose, and Rhamnose on NHDFs Viability

The cytotoxicity of rhamnose, rutinose, hesperidin, and hesperetin on primary NHDFs was evaluated using the MTT test. Rhamnose and rutinose were tested in the concentration range of 1–100 mM and hesperidin and hesperetin in the concentration range of 1–100 µM. Concentrations of 5–100 mM for rhamnose and 5–75 mM for rutinose significantly reduced the viability of NHDFs. For hesperidin, a significant reduction was observed at 100 µM concentration, and hesperetin did not reduce the viability of NHDFs in the tested range. As a control, untreated NHDFs cultured in a serum-free medium containing DMSO (0.1%) were used.

Based on the MTT test results, two concentrations of studied compounds were selected for further experiments: rhamnose (1 and 10 mM, where the viability of NHDFs was 88 and 78%, respectively), rutinose (1 and 10 mM, where the viability of NHDFs was 78 and 62%, respectively), and for hesperidin and hesperetin nontoxic concentration (1 and 10 µM) was chosen (Figure 3).

### 2.3. Effect of Hesperidin, Hesperetin, Rutinose, and Rhamnose on Skin-Aging Models

Aging also leads to the overexpression of MMPs as well as a decrease in collagen synthesis in the ECM. Aging is also associated with the production of inflammatory mediators. The effect of rutinose, rhamnose, hesperidin, and hesperetin on the production of MMPs in young and physiologically aged NHDFs was investigated.

The expression of MMP-1 was reduced in both young and physiologically aged NHDFs after administration of the tested compounds. Rutinose and rhamnose (10 mM) and hesperetin (10 µM) decreased the expression of MMP-1 in young NHDFs by approximately 15% (Figure 4A). Hesperidin (10 µM) and hesperetin (10 µM) treatment also caused a decrease in MMP-1 level (~20%) in physiologically aged NHDFs (Figure 4B).

MMP-2 levels were significantly decreased in young NHDFs (low-glucose model) after the application of 10 mM rutinose (by 30%), rhamnose (by 20%), and 10 µM hesperetin (by 15%). In glycated models, the decrease in MMP-2 level was slightly lower after the application of rutinose, rhamnose, and hesperidin (Figure 4C). In physiologically aged NHDFs (high-glucose and AGE models), a decrease in MMP-2 level was observed with all tested compounds. A significant decrease was observed after treatment with 10 mM rutinose in the low-glucose model (Figure 4D).

The production of collagen I has increased in young NHDFs after the application of carbohydrates—rhamnose and rutinose. No effect was observed after the administration of flavonoids, e.g., hesperidin and hesperetin in the low-glucose model, and a decrease in collagen I production was observed in the glycated model (Figure 4E). In physiologically aged NHDFs, collagen I production was only slightly increased after the treatment with rhamnose and rutinose and decreased after treatment with hesperidin and hesperetin. Hesperetin (10 µM) treatment significantly decreased collagen I production to 40–50% compared to control cells (Figure 4F).

Furthermore, the anti-inflammatory effect of rhamnose, rutinose, hesperidin, and hesperetin in skin-aging models, quantified as the levels of IL-6 and IL-8, which are overexpressed in senescent cells, were measured.

In the young NHDFs (low-glucose model), the IL-6 level was reduced by almost all tested compounds (except for 10 mM rutinose and 1 µM hesperetin). The results in the high-glucose model were comparable to the control or slightly increased in the AGE-induced aging model, where only rhamnose reduced IL-6 concentration by 10% (Figure 5A). In physiologically aged NHDFs, IL-6 levels decreased by ca. 10% when treated with rhamnose and were comparable to the control after treatment with rutinose. However, we observed the greatest effect after treatment with 10 μM hesperidin and hesperetin. In the low-glucose model, hesperidin reduced IL-6 level by 20%, and hesperetin by up to 30%. Hesperetin was similarly effective in both glycated models (high-glucose and AGEs) (Figure 5B).

In the young NHDFs (low-glucose model), the level of IL-8 was lowered mainly by 10 mM rhamnose and 10 mM rutinose. In the high-glucose model, the level of IL-8 was also decreased by approximately 20% after treatment with rutinose, hesperidin, and hesperetin. In contrast, the levels of IL-8 were increased after treatment with the tested compounds in the AGE-stimulated NHDFs; only rhamnose decreased the IL-8 concentration by 20% (Figure 5C). In physiologically aged NHDFs, the level of IL-8 was reduced by approximately 25% after treatment with 10 mM rhamnose. When cells were treated with other compounds, it was slightly increased, and with 1 mM rutinose, it was comparable to the control (Figure 5D).

## 3. Discussion

Skin aging is characterized by thinning of the epidermis, loss of hydration and degradation of collagen and elastic fibers, by the accumulation of modified (glycated and oxidized, hydrolytic products) components of the ECM and by slowing the cellular regeneration process. Skin aging is caused by internal factors, but it is also significantly influenced by external factors from the environment (pollution, UV light, etc.) and lifestyle (e.g., smoking, alcohol abuse, or poor nutrition) [18]. The internal factors that modulate skin aging include time, genetic factors, and hormones. They lead not only to the accumulation of cellular damage and degradation of ECM components but also to the production of reactive oxygen species (ROS) and AGEs [4,16,18]. Increased accumulation of AGEs in human tissues has been associated with end-stage renal disease, chronic obstructive pulmonary disease, and more recently, skin aging [19]. AGEs were also shown to increase the expression of elastase-type MMPs and endopeptidases when added to NHDFs [20].

One of the ways to influence the molecular pathways of aging is to use natural compounds (polyphenols) with anti-oxidant and anti-glycation properties, which delay the aging process and subsequently alleviate age-related skin pathologies [1]. In our study, the anti-aging potential of the flavonoids hesperidin and hesperetin and their carbohydrate components rutinose, and rhamnose were investigated. The transdermal distribution of hesperidin has already been demonstrated, so we focused on its biological activity during skin aging [11]. First, their ability to inhibit the activities of enzymes associated with skin aging, specifically hyaluronidase, collagenase, and elastase, which degrade the components of the ECM, was determined in the tube test. Collagenase and elastase are proteolytic enzymes directly involved in the degradation of collagen and elastin present in the ECM. Excessive expression of these enzymes, as well as hyaluronidase in skin tissue, causes the formation of wrinkles, sagging skin, loss of skin elasticity, and a decrease in skin hydration [21,22]. In the tube tests, rutinose was found to be the most effective inhibitor of elastase, hyaluronidase, and collagenase activity among all tested compounds in the tested concentration range, and compared with standard inhibitors. It inhibited elastase activity to an extent similar to its standard inhibitor oleanolic acid.

In the case of one structural part of rutinose–rhamnose, the highest effect was found in the inhibition of hyaluronidase. The effect of rhamnose on the inhibition of elastase and collagenase was also observed. However, compared with rutinose, rhamnose had about half the inhibitory effect on these enzymes. The effect of rhamnose on the inhibition of elastase was previously published in a study by Andrès et al. (2006), which demonstrated that rhamnose-rich oligo- and polysaccharides stimulate cell proliferation, inhibit elastase activity, stimulate collagen biosynthesis, and protect hyaluronan from free radical-mediated degradation [23]. Another study correlated with the stimulation of collagen production published by Pageon et al. (2019) shows that rhamnose stimulates the production of procollagen I in the superficial dermis, which was observed in both in vitro and in vivo models [13]. The mode of action of rhamnose is not clear, but several reports corroborate the hypothesis that rhamnose may interact with a specific lectin-type receptor expressed on the surface of fibroblasts [24].

As for hesperidin and hesperetin, they, as well as rhamnose, showed the greatest effect in inhibiting hyaluronidase. They also weakly inhibited elastase activity, with hesperidin being slightly stronger than hesperetin, which is consistent with a study by Vidhya et al. (2022), where hesperidin also inhibited human neutrophil elastase [25]. Its effect on the inhibition of collagenase activity was not detected. However, in the study by Stanisic et al. (2020), inhibition of collagenase by the addition of hesperidin was observed, though at higher concentrations (0.08–0.9 mM) than we tested in our study [8]. The results of a comparative study on flavonoids published by Li et al. (2021) showed a high binding constant of hesperidin to hyaluronidase (binding constant was 3.740 × 10^4^ L mol^−1^ at 293 K), which may lead to a decrease in enzyme activity. This binding constant was even higher than that of myricetin, puerarin, genistein, and naringin [26]. The application of hesperidin in the clinical follow-up study by Sheen et al. (2021) was also related to the protection of hyaluronan. It was reported that a topical formulation containing 0.1% hesperetin applied twice daily for 12 weeks increased *stratum corneum* hydration, skin elasticity, and HA accumulation in the dermis [10]. Topical application of hesperidin was described in a study by Man et al. (2015), in which they demonstrated that topically applied hesperidin improved epidermal permeability barrier homeostasis while lowering skin surface pH in aging mouse skin [27].

In addition, the cytotoxicity of the investigated compounds was determined using the MTT assay on NHDFs. For the topical application of the tested substances, it is necessary to use their sub-toxic concentrations. The concentrations of 100 mM rhamnose and rutinose and 100 μM hesperidin demonstrated a toxic effect on cell viability. This is consistent with Roohbakhsh et al. (2015) who demonstrated toxicity in NALM-6 cells after the application of hesperidin and hesperetin [28]. Subtoxic concentrations of hesperidin, hesperetin (1–25 μM), rhamnose, and rutinose (1–25 mM) were tested in skin aging models. The antisenescence activity of the tested compounds was evaluated by measuring MMP-1 and MMP-2 levels. Both MMPs studied are calcium-dependent zinc endopeptidases that play a crucial role in both extrinsic and intrinsic aging processes. These enzymes are responsible for ECM degradation and tissue remodeling [29].

The levels of MMP-1 and MMP-2 were reduced in both young NHDFs and physiologically old NHDFs after the application of the test substances. Hesperidin and hesperetin (10 µM) decreased MMP-1 levels in physiologically aged NHDFs by approximately 20%, which is consistent with the study by Santhanam et al. [21], in which hesperidin also decreased the levels of MMP-1 as well as MMP-3 and MMP-9 in UVB-irradiated human fibroblasts. Additionally, the study by Lu et al. (2022) showed that pretreatment with hesperetin suppressed the UVA-induced inflammatory response by downregulating MMP-1, IL-6, and cyclooxygenase 2 [30].

The most promising candidate for reducing MMP-2 levels was 10 mM rutinose, which significantly decreased the level of this enzyme in both young NHDFs and physiologically aged NHDFs. Moreover, all tested compounds were more effective in glycated models (induced by high glucose and AGEs) with physiologically aged NHDFs, demonstrating their protective potential against glycation. Hesperidin also proved to be a potent anti-photoaging factor by regulating metalloproteinases via mitogen-activated protein kinase (MAPK) signaling pathways. In the same study, Lee et al. (2018) confirmed the beneficial effect of hesperidin on wrinkle depth in a mouse dorsal skin model [9].

In the production of collagen I, we observed an increased production in NHDFs after the application of the tested compounds—rhamnose and rutinose. In the case of the flavonoids hesperidin and hesperetin, the results were comparable to the control. On the other hand, Hiraishi et al. (2017) reported that hesperidin improved the mechanical properties of the collagen matrix and its resistance to biodegradation [31]. A positive effect on the upregulation of collagen production was also observed in the study by Péterszegi et al. (2008), in which rhamnose-rich polysaccharides caused a significant upregulation of collagen deposition [32].

Last but not least, we focused on the possible anti-inflammatory effect of the tested substances by measuring IL-6 and IL-8. Here, it was strongly dependent on the aging model (low or high glucose). In young NHDFs (low glucose model), IL-6 was reduced by almost all tested substances. In glycated models, only rhamnose decreased the level of IL-6. In physiologically aged NHDFs, the IL-6 level was also reduced by treatment with rhamnose, but we observed the greatest effect after treatment with 10 μM hesperidin and hesperetin. In young NHDFs (low-glucose model), the levels of IL-8 were decreased mainly by rhamnose and rutinose. In the glycated model, the level of IL-8 also decreased after treatment with hesperidin and hesperetin (high glucose model). In physiologically aged NHDFs, the level of IL-8 decreased mainly after rhamnose treatment. The fact that hesperidin and hesperetin can regulate oxidative stress and inflammatory activity by down-regulating IL-1β, IL-6, and tumor necrosis factor α (TNF-α), has already been published in a number of studies on skin cells exposed to UVA or UVB radiation [30,33,34]. Huang et al. (2018) postulated that hesperetin decreased the secretion of IL-6 by infected RAW 264.7 [35]. In addition, hesperidin improved epidermal barrier function and reduced the level of pro-inflammatory cytokines TNF-α, IL-1, and IL-6 in the lipopolysaccharide-induced mouse model [36].

## 4. Materials and Methods

### 4.1. Chemicals

#### 4.1.1. Rhamnose, Rutinose, Hesperidin, and Hesperetin

l-Rhamnose, hesperidin, and hesperetin were purchased from Merck (Waltham, MA, USA). Rutinose was prepared by the enzymatic cleavage of rutin Merck (Waltham, MA, USA) with rutinosidase from *Aspergillus niger* K2 according to the published procedure [37].

#### 4.1.2. Other Chemicals

Heat-inactivated fetal bovine serum (FBS, HyCylone™) and PBS (10X), pH 7.4 were purchased from Thermo Fisher Scientific (Waltham, MA, USA). Dulbecco’s modified Eagle’s medium high glucose 4.5 g/L (DMEM), penicillin-streptomycin solution, trypsin-EDTA (0.25%), 3-(4,5-dimethylthiazol-2-yl)-2,5-diphenyltetrazolium bromide (MTT), 2,2′-azino-bis(3-ethylbenzothiazoline-6-sulfonic acid) (ABTS substrate), 3,3′,5,5′-tetramethylbenzidine (TMB substrate), human glycated albumin, elastase from human leukocytes, *N*-succinyl-Ala-Ala-Ala-*p*-nitroanilide, oleanolic acid, hyaluronidase from bovine testes, 4 (dimethylamino)benzaldehyde (DMAB), hyaluronic acid sodium salt from cockscomb, sodium aurothiomalate hydrate (SATMH) and Collagenase Activity Colorimetric Assay Kit (MAK293) were purchased from Merck (Waltham, MA, USA). ELISA Human IL-6 Kit (900-K16) and ELISA Human IL-8 Kit (900-K18) were purchased from PeproTech (London, UK). Human Total MMP-1 DuoSet ELISA (DY901B-05), Human MMP-2 Duo-Set ELISA (DY902), and Human Pro-Collagen I alpha 1 DuoSet ELISA (DY6220-05) were purchased from R&D Systems (Minneapolis, MI, USA).

### 4.2. Determination of the Activity of Enzymes Associated with Skin Aging by Tube Tests

#### 4.2.1. Elastase Activity

The anti-elastase activity was determined using a method by Vostálová et al. [1]. The reaction mixture contained 100 μL of 0.1 M HEPES buffer (pH 7.5), 10 μL of the test sample (hesperidin, hesperetin, rutinose, or rhamnose)/inhibitor (oleanolic acid in DMSO; 1.46 mg/mL) or solvent DMSO, and 20 μL of elastase enzyme (1 U/mL) except for the blank. All tubes were incubated at room temperature for 5 min, the reaction was initiated by adding 30 μL of the substrate, *N*-succinyl-Ala-Ala-Ala-*p*-nitroanilide (4.4 mM). The reaction was monitored as the change in absorbance (A) at 410 nm (∆A/min) using a UV-VIS recording spectrophotometer (Shimadzu (Kyoto, Japan); UV-2401PC) and the percentage inhibition was calculated according to Equation (1):Inhibition [%] = 100 − [(A_test_ sample − Ablank)**/**(A_control_ − A_blank_)] × 100(1)

#### 4.2.2. Hyaluronidase Activity

Anti-hyaluronidase activity was determined according to the method by Ndlovu et al. with minor modifications [38]. First, 25 μL CaCl_2_ (12.5 mM) and 12.5 μL of the tested samples (hesperidin, hesperetin, rutinose, or rhamnose) dissolved in DMSO were mixed in the test tubes. All tubes except the blank contained 12.5 μL of the enzyme hyaluronidase (1.5 mg/mL). The blank contained 12.5 μL DMSO and 12.5 μL distilled water instead of the enzyme. Sodium aurothiomalate (2.8 mg/mL) was used as the inhibitor of hyaluronidase. After incubation at 37 °C for 20 min, the reaction was started by adding 100 μL of the substrate hyaluronic acid (1 mg/mL). After incubation (40 min), the reaction was terminated by the addition of 25 μL KBO_2_ (0.8 M), and the tubes were incubated at 100 °C for 3 min and cooled down to room temperature. After the addition of DMAB (800 μL, 4 g DMAB in 40 mL CH_3_COOH and 5 mL 10 M HCl), the tubes were incubated for 20 min at room temperature. The reaction mixture (200 μL) was transferred into a 96-well plate, and fluorescence was detected at 545 nm excitation and 612 nm emission. Percent inhibition was calculated according to Equation 1.

#### 4.2.3. Collagenase Activity

To determine the effect of the studied compounds on collagenase activity in the tube assay, we used the Collagenase Activity Colorimetric Assay Kit. Briefly, all samples and standards (blank; positive control; inhibitor –1,10-phenanthroline (1 M)) were mixed with collagenase assay buffer and collagenase (0.35 U/mL) at the desired concentration in a total volume of 100 µL/well. For each reaction, a 100 µL/well reaction mixture was prepared and supplemented with 40 µL of collagenase substrate (FALGPA; *N*-(3-[2-furyl]acryloyl)-Leu-Gly-Pro-Ala) mimicking collagen structure and 60 µL of collagenase assay buffer. Absorbance was measured at 345 nm in a microplate reader at 37 °C for 5–90 min in the kinetic mode. The percentage inhibition was calculated using Equation (1).

### 4.3. Cell Culture and Preparation of Skin-Aging Models

NHDFs were isolated from skin fragments [39]. The skin specimens were obtained from healthy tissue taken from plastic surgery patients from Faculty Hospital Olomouc after informed consent. The study was performed according to the Code of Ethics of the World Medical Association (Ref. number 41/09; Ethics Committee of the University Hospital and Faculty of Medicine Palacký University Olomouc; Date: 6 April 2009 in Olomouc (CZ)).

The cells between passages 2–3 were used for the experiments with young NHDFs and the cells between passages 15–16 were used for the experiments with physiologically aged NHDFs. The morphology of NHDFs was microscopically evaluated during the entire period of cultivation. NHDFs were cultured in low glucose (25 mM glucose—physiological aging model) and high glucose (50 mM glucose—accelerated aging model) DMEM medium supplemented with FBS (10%; *v/v*) and antibiotics (penicillin (100 mg/mL) and streptomycin (100 U/mL)) at 37 °C in 5% CO_2_ humidified atmosphere. The medium was exchanged every 72 h. In the last aging model (AGEs-stimulated aging), NHDFs were cultured in a low-glucose medium, and 24 h before the experiments the low-glucose medium was replaced with a medium supplemented with human glycated albumin (200 µg/mL).

### 4.4. Cell Viability Assay

NHDFs were seeded in 96-well plates at a final density of 0.2 × 10^4^ cells/well. On the following day, the tested compounds: hesperidin, hesperetin (both 1 to 100 µM), rutinose or rhamnose (both 1 to 100 mM) in serum-free DMEM were added to the cells and incubated for 24 h. Cell viability was assessed using an MTT assay, based on the ability of mitochondrial enzymes of living cells to metabolize yellow water-soluble tetrazolium salt into insoluble purple formazan, which was dissolved in DMSO with NH_3_ (1.0%, *v/v*) and quantified spectrophotometrically at 540 nm [40].

### 4.5. Effect of Studied Compounds on Skin-Aging Models

NHDFs between passages 2–3 (young NHDFs), between passages 15–16 (physiologically aged NHDFs) and NHDFs stimulated with human glycated albumin were seeded in 6-well plates and cultured until confluence. Freshly prepared solutions of the tested substances were applied to the cells. Stock solutions of hesperidin and hesperetin (10 mM; DMSO) were diluted in serum-free medium to a final concentration of 1 and 10 µM, respectively. Rutinose and rhamnose were dissolved in DMSO (20 mM stock solutions) and diluted in a serum-free medium to a final concentration of 1 and 10 mM, respectively. The total DMSO content was 0.1% (*v/v*). Test samples were applied to the individual in vitro skin models (young NHDFs, physiologically aged NHDFs and accelerated aged NHDFs) for 24 h, then the cell culture medium was collected and stored at –80 °C until analysis.

### 4.6. Evaluation of the Effect of Hesperidin, Hesperetin, Rutinose, and Rhamnose on Skin Aging Models

Determination of matrix metalloproteinases (MMP-1, MMP-2), collagen I, IL-6, and IL-8 level.

The levels of MMP-1, MMP-2, collagen I, IL-6, and IL-8 after treatment with sub-toxic concentrations of hesperidin and hesperetin (both 1 and 10 µM), rutinose and rhamnose (both 1 and 10 mM) in the three different aging models were examined in cell culture medium using the commercial kits (Human Total MMP-1 DuoSet ELISA (DY901B-05), Human MMP-2 DuoSet ELISA (DY902), Human Pro Collagen I alpha 1 DuoSet ELISA (DY6220-05), ELISA Human IL-6 Kit (900-K16) and ELISA Human IL-8 Kit (900-K18) according to the manufacturer’s instructions.

### 4.7. Statistical Analysis

The results of enzyme activity measurements are expressed as the mean of 3 measurements ± SD. The results of in vitro measurements on individual skin aging models are expressed as the mean of 6 measurements ± SD. Values of * *p* < 0.05; ** *p* < 0.01; *** *p* < 0.001; **** *p* < 0.0001 were considered significant compared to Control (C). (One-way ANOVA; Dunnett’s multiple comparison test was performed using GraphPad Prism).

## 5. Conclusions

Our study demonstrated the anti-aging potential of the flavonoids hesperidin and hesperetin and their carbohydrate portions rutinose, and rhamnose. Rutinose and rhamnose inhibited the activity of isolated elastase, hyaluronidase, and collagenase. Hesperidin and hesperetin inhibited elastase and hyaluronidase activity. Most importantly, the tested compounds, used in subtoxic concentrations, were all able to act as antiglycation agents, which is important not only in the prevention of skin aging but also for the prevention of the development of civilization’s diseases. In skin-aging models, they also reduced the levels of pro-inflammatory interleukins and matrix metalloproteinases. Our study suggests that rhamnose and especially rutinose may have potential applications in cosmetic and dermatological preparations with anti-wrinkle activity.

## Figures and Tables

**Figure 1 molecules-28-01728-f001:**
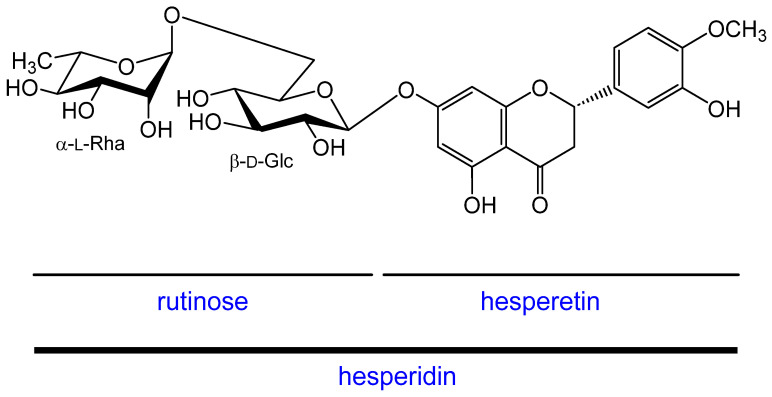
Structure of hesperidin.

**Figure 2 molecules-28-01728-f002:**
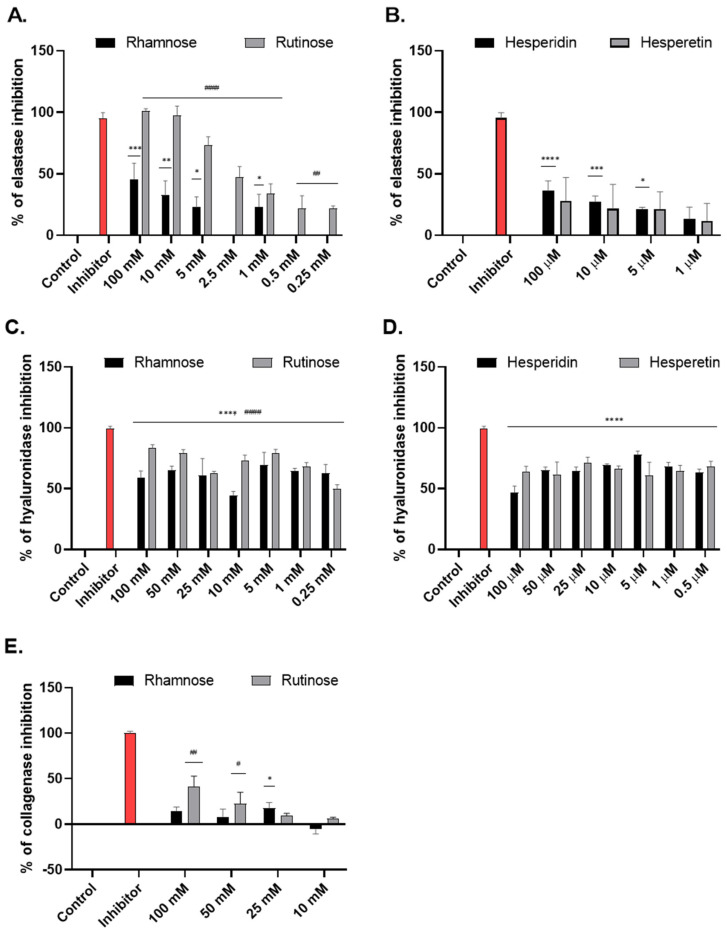
The effect of rhamnose, rutinose, hesperetin, and hesperidin on the pure enzymes: elastase (**A**,**B**), hyaluronidase (**C**,**D**), and collagenase I (**E**) activity using tube tests. Oleanolic acid (1.46 mg/mL) was used as the inhibitor of elastase (red color), sodium aurothiomalate (2.8 mg/mL) was used as the inhibitor of hyaluronidase (red color) and 1,10-phenanthroline (1 M) was used as the inhibitor of collagenase (red color). The results are expressed as an average of 3 measurements. * *p* < 0.05; ** *p* < 0.01; *** *p* < 0.001; **** *p* < 0.0001 were considered significant compared to control C. ^#^
*p* < 0.05; ^##^
*p* < 0.01; ^####^
*p* < 0.0001 were considered significant compared to control (for rutinose). (One-way ANOVA; Dunnett’s multiple comparisons tests were performed using GraphPad Prism).

**Figure 3 molecules-28-01728-f003:**
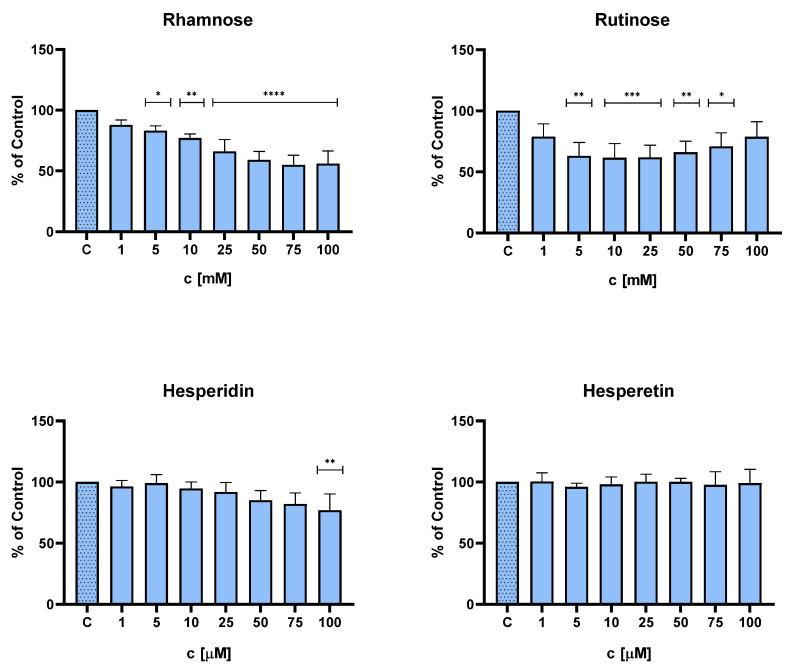
Effect of rhamnose, rutinose, hesperidin, and hesperetin on NHDFs viability. Untreated NHDFs cultured in a serum-free medium containing DMSO (0.1%; *v/v*) were used as a negative control (C). Results are expressed as a percentage of control viability and SD. Number of measurements: *n* = 4. * *p* < 0.05; ** *p* < 0.01; *** *p* < 0.001; **** *p* < 0.0001 were considered significant compared to control. (One-way ANOVA; Dunnett’s multiple comparisons tests were performed using GraphPad Prism).

**Figure 4 molecules-28-01728-f004:**
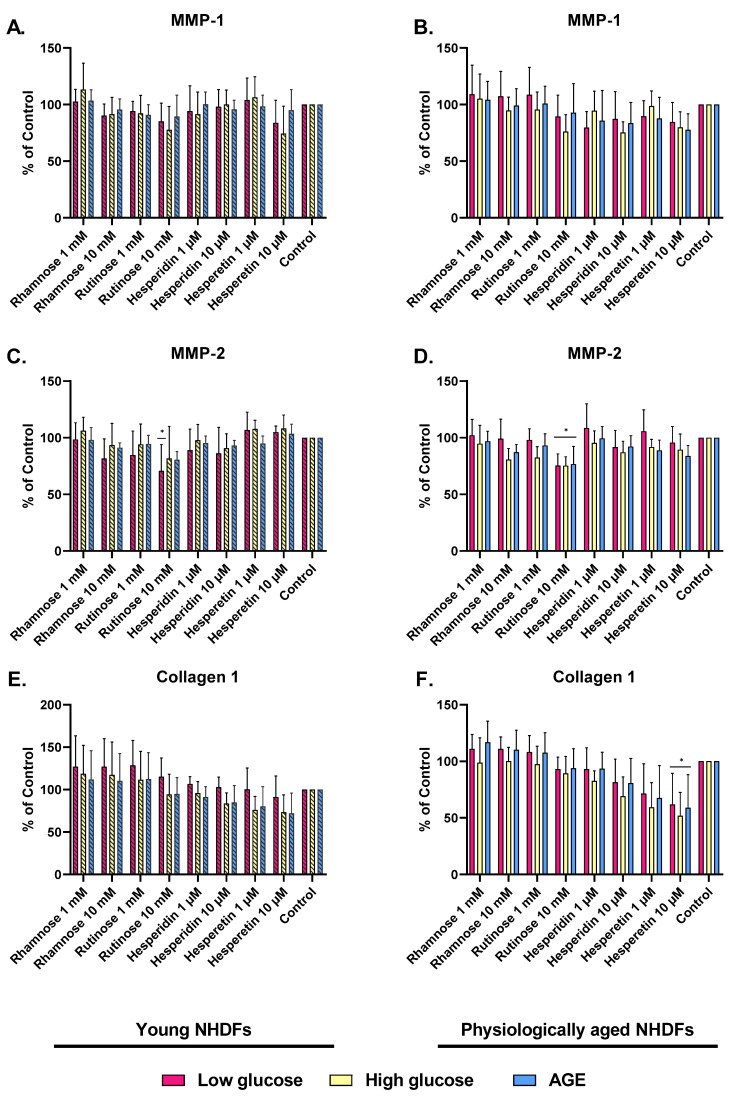
The effects of rhamnose, rutinose, hesperidin, and hesperetin on MMP-1, MMP-2, and collagen 1 production. MMP-1 (**A**), MMP-2 (**C**), and collagen 1 (**E**) production in young NHDFs, that were cultivated in low glucose medium or high glucose medium or medium supplemented with AGEs and the effects of rhamnose, rutinose, hesperidin, and hesperetin on MMP-1 (**B**), MMP-2 (**D**) and collagen 1 (**F**) production in physiologically aged NHDFs cultivated in low-glucose medium, high-glucose medium and medium supplemented with AGEs. Untreated NHDFs cultured in the respective media were used as controls. A number of measurements: *n* = 6. * *p* < 0.05 were considered significant compared to the control. (One-way ANOVA; Dunnett’s multiple comparisons tests were performed using GraphPad Prism).

**Figure 5 molecules-28-01728-f005:**
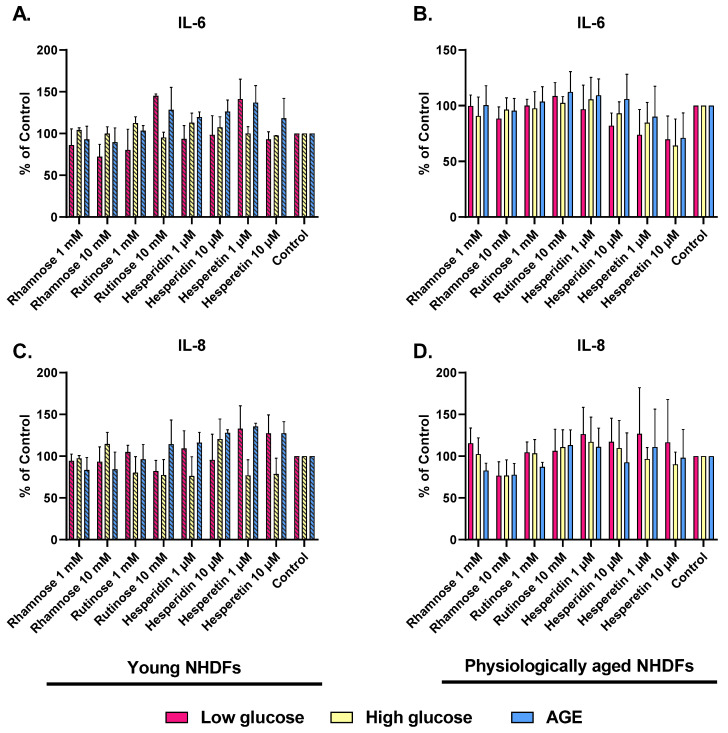
The effects of rhamnose, rutinose, hesperidin, and hesperetin on IL-6 (**A**,**B**) and IL-8 (**C**,**D**) production by young or physiologically aged NHDFs, that were cultivated in low-glucose medium, high-glucose medium, and medium supplemented with AGEs. Untreated NHDFs cultivated in these individual media were used as controls. A number of measurements: *n* = 6. (One-way ANOVA; Dunnett’s multiple comparisons tests were performed using GraphPad Prism).

## Data Availability

Data are available upon request from the corresponding author.

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
