# Peer review of "Hesperidin, Hesperetin, Rutinose, and Rhamnose Act as Skin Anti-Aging Agents"

_molecules, 2023, doi:10.3390/molecules28041728_

Round 1
Reviewer 1 Report
Thank you for your efforts. I think that the results expressed within the article are far away from applied science. Of course, we believe the invitro assays. But we should consider also the bioavailability of the tested compounds. It is well known that flavonoid glycosides are highly polar and they are of very low absorption capacity. One more important notice, about the activity of Rhamnose, Rhamnose exerted a very potent activity. It is a sugar. What is next? we could give it as a medicine. I suggest that the authors remove the activities of the sugars and keep only the flavonoid glycosides. Moreover, please study the pharmacokinetic activities of the studied compounds and include them within the article. I am also worrying about the interference of the yellow color (flavonoids are yellow colored compounds) with the measurement in the invitro assays.
Author Response
Dear Reviewer,
Thank you for reading our manuscript and giving us the opportunity to improve it. We have revised the manuscript based on your comments and highlighted the changes in the manuscript in yellow (Please see the attachment).
Information about the bioavailability and possible transdermal transport of hesperidin has been added to the manuscript.
Thank you for your comment on the application of flavonoids and sugars. In order to achieve a biological effect for possible topical application, we tested substances at different subtoxic concentrations, since sugars have a lower biological response. We consider it extremely important to test rhamnose as a component (moiety) of the tested flavonoid substances both in terms of specific activity and also toxicity. Moreover, rhamnose is already used in some specific dermatological anti-wrinkle preparations (e.g., l’Oreal or Vichy Laboratories, https://www.vichyusa.com/blog/skin-concern/wrinkles-fine-lines/top-5-anti-aging-ingredients.html), so yes – rhamnose is already used for some dermatological treatments. Moreover, we have shown that rhamnose and rutinose at these subtoxic concentrations have great antiglycation potential and reduce enzyme activities associated with skin aging. The next step would be to verify this mechanism in clinical trials.
Regarding the interference of the yellow color of flavonoids. We used low concentrations in the in vitro tests where this coloration was completely insignificant, and in addition, a blank test was used in the measurement for all concentrations used to address this (potential) problem.
Thank you once again for your kind review.
Reviewer 2 Report
This manuscript presents the study in human dermal fibroblasts (NHDFs) induced by physiological or by high glucose or by advanced glycated end products (AGEs) of four substances: hesperidin and its main sub-unities: hesperetin, rutinose and rhamnose, as antiglycating agents targeting promising candidates to act as a preventive strategy of aging. Matrix metalloproteinases (collagenase and gelatinase A, that were reduced by all substances) and inflammatory interleukins (ILs) by ELISA were analysed together with elastase, hyaluronidase, and collagenase activities. Rutinose and rhamnose inhibited the activity of pure elastase, hyaluronidase, and collagenase. Hesperidin and hesperetin inhibited only elastase and hyaluronidase activity.
At the title and abstract, it would be more clear describe that it is a study of hesperidin and its main sub-unities: hesperetin, rutinose and rhamnose. Abstract should also bring more quantitative results. The results obtained to flavonoids and sugars were quite different, but undiscussed.
References are out of order. Reference 15 is at introduction, line 79, but reference 16 is at line 316, experimental.
More references are needed at discussion section.
Experimental section is well-formulated with recent published references.
Some references are out of format. They are updated and no self-citation issue was observed.
Author Response
Dear Reviewer,
Thank you for reading our manuscript and giving us the opportunity to improve it. We have revised the manuscript based on your comments and changes were highlighted in the manuscript in yellow color (Please see the attachment).
It was specified in the abstract that this is a study of hesperidin and its main subunits: hesperetin, rutinose, and rhamnose. We also added more results in the abstract.
Thank you for your comment regarding the discussion of the flavonoids and sugars results. In order to achieve a biological effect for a possible topical application, we tested substances in different sub-toxic concentrations, since sugars have a lower biological response. This is the reason why the discussion is conceived in this way.
Thank you for pointing out the references. They have all been corrected and the format of all references has been changed to match the style of the journal.
Reviewer 3 Report
The manuscript entitled: Hesperidin, Hesperetin, Rutinose, and Rhamnose as Skin Anti- 2 Aging Agents; presents an interesting study which concluded that rhamnose may have potential applications in cosmetic and dermatological preparations with anti-wrinkle activity. Yet the following points are to be addressed before further steps:
Abstract
-Rewrite line 25
Introduction
-Line 57: (Therefore, it was selected as a reference compound). Is Hesperidin a reference of test compound?
Results
-It is important to rethink the use of the word (Only) all over the manuscript.
-Figure 2: Please explain why the x-axis starts with the highest concentration?
Discussion is rich and well written.
Materials and methods
-More information needed about the informed consent (where it took place, procedure, authorizing body, date and number).
-Line 385: do you mean 104?
Author Response
Dear Reviewer,
Thank you for reading our manuscript and giving us the opportunity to improve it. We have revised the manuscript based on your comments and changes were highlighted in the manuscript in yellow color (Please see the attachment).
Specifically:
Abstract
Line 25 was rewritten. Line 57 has been removed. We agree with you that hesperidin is not the reference substance. In our research, this is the most discussed substance, so we have labeled it as such.
Results
The word "only" was corrected in the manuscript.
Figure 2; x-axis: this is a graphical representation only. We agree that it could also be expressed from the lowest to the highest concentration. We proceeded by determining inhibition from more concentrated substances to lower concentrations that can still cause inhibition in determining inhibition. So, this is the reason for our graphical representation.
Materials and methods
Thank you for your comment, we have added information about informed consent to the manuscript.
Moreover, minor typos were corrected throughout the paper.
Round 2
Reviewer 1 Report
The article can be accepted in its current form